# Mixture-of-Experts with Gradient Conflict-Driven Subspace Topology Pruning for Emergent Modularity

## Abstract

Mixture-of-Experts (MoE) architectures achieve parameter efficiency through conditional computation, yet contemporary designs suffer from structural parameter isolation. We propose CDSP-MoE (Conflict-Driven Subspace Pruning MoE), a framework that aims to alleviate these structural bottlenecks through a paradigm shift from isolated expert containers to *dynamic expert instantiation within a shared physical subspace*. Grounded in the Universal Weight Subspace Hypothesis, CDSP-MoE maintains a super-complete parameter backbone where logical experts are carved out via learnable topology masks. Unlike prior work that uses gradient conflict for token reassignment or optimization surgery, we leverage it as a *structural supervisory signal*: a Lagged Gradient Game penalizes interfering connections in the shared manifold, enabling the topology to spontaneously prune conflicting pathways and evolve interpretable modular structures. Experimental results demonstrate that CDSP-MoE achieves robust content-driven routing without human-defined task labels, maintaining semantic specialization even under strict blind inference protocols where explicit instructions are absent.

**Keywords:** Mixture-of-Experts, Gradient Conflict, Subspace Topology Pruning, Dynamic Expert Instantiation, Emergent Modularity, Instruction-Free Routing, Shared Parameter Manifold, Lagged Gradient Game, Structural Evolution, Multi-Task Learning.

[1]Anonymous Institution, Anonymous City, Anonymous Region, Anonymous Country. Correspondence to: Anonymous Author <anon.email@domain.com>.

Preliminary work. Under review by the International Conference on Machine Learning (ICML). Do not distribute.

## 1. Introduction

The scaling of large language models has been driven by the Mixture-of-Experts (MoE) paradigm. By activating only a subset of parameters per token, systems such as GShard (Lepikhin et al., 2021), Switch Transformer (Fedus et al., 2022), and GLaM (Du et al., 2022) achieve efficient scaling under unified routing and scaling laws (Clark et al., 2022; Zoph et al., 2022). Recent architectures, including DeepSeek-MoE (Dai et al., 2024), further improve this paradigm via fine-grained expert segmentation and partial sharing.

Despite this progress, standard MoE relies on *structurally isolated* experts. Since experts are disjoint tensors, routers must use auxiliary losses to enforce load balancing (Fedus et al., 2022; Zhou et al., 2022; Hazimeh et al., 2021; Riquelme et al., 2021). This creates two coupled failure modes. First, forced balancing routes unrelated tokens through the same expert, inducing gradient interference and forgetting (Kirkpatrick et al., 2017; Lopez-Paz & Ranzato, 2017; Chaudhry et al., 2019; Sener & Koltun, 2018; Kendall et al., 2018; Zenke et al., 2017; Aljundi et al., 2018; Farajtabar et al., 2020). Second, strict expert isolation fragments representations, limiting compositionality and robustness.

Such effects are manifestations of gradient conflict in shared parameter spaces (Yu et al., 2020; Ruder, 2017; Yang et al., 2025). While most methods treat conflict as noise to be suppressed, in MoE it reflects task-level incompatibility and latent structure.

We propose Mixture-of-Experts with Conflict-Driven Subspace Pruning (CDSP-MoE). Our approach is motivated by the Universal Weight Subspace Hypothesis (Kaushik et al., 2025), and by evidence that neural solutions lie in low-dimensional, connected manifolds (Frankle & Carbin, 2019; Garipov et al., 2018; Izmailov et al., 2018; Draxler et al., 2018; Wortsman et al., 2021; 2022). LoRA-style adaptations (Hu et al., 2022; Zhang et al., 2023) provide a practical instantiation of this view. CDSP-MoE extends it by dynamically carving experts from a shared physical backbone.

At the core of CDSP-MoE is a *Lagged Gradient Game* that uses gradient cosine similarity to penalize interfering

connections. This induces topology evolution in a shared subspace, yielding emergent modular experts without pre-defined boundaries. Related perspectives on architecture and routing discovery appear in routing networks and task grouping (Rosenbaum et al., 2018; Standley et al., 2020; Ruder et al., 2019).

To study these dynamics in a controlled setting, we evaluate CDSP-MoE on heterogeneous multi-task benchmarks. The resulting behavior is consistent with classical theories of stochastic processes and high-dimensional concentration (Ledoux, 2001; Li et al., 2017; Doob, 1953; Risken, 1996; Diaconis & Freedman, 1984; Vershynin, 2018).

## 2. Related Work

### 2.1. Sparse Mixture-of-Experts

Conditional computation was introduced in the sparsely-gated MoE layer (Shazeer et al., 2017), which uses noisy top-$k$ routing to activate a small subset of experts per token. This paradigm was scaled by systems such as GShard (Lepikhin et al., 2021), Switch Transformer (Fedus et al., 2022), GLaM (Du et al., 2022), and ST-MoE (Zoph et al., 2022), which focus on efficiency, stability, and large-scale training. Theoretical scaling behavior for routed models was later analyzed in (Clark et al., 2022). More recently, DeepSeek-MoE (Dai et al., 2024) introduced fine-grained expert segmentation and shared experts to improve specialization and utilization.

**Routing and Expert Selection.** To mitigate load imbalance in top-$k$ routing, Expert Choice Routing (Zhou et al., 2022) allows experts to select tokens under capacity constraints. Differentiable selectors such as DSelect-k (Hazimeh et al., 2021) further formulate routing as a continuous optimization problem. Large-scale vision MoE models also exhibit emergent specialization without strict partitions (Riquelme et al., 2021). These approaches focus on assigning tokens to experts under a fixed parameterization. In contrast, CDSP-MoE operates on a shared parameter space, where conflict-driven pruning determines which physical dimensions constitute each expert.

### 2.2. Subspace Learning and Low-Rank Adaptation

Our approach is grounded in the Universal Weight Subspace Hypothesis (Kaushik et al., 2025), which posits that diverse tasks can be represented by combinations of a shared low-dimensional basis. Related evidence comes from work on sparse subnetworks and connected solution manifolds (Frankle & Carbin, 2019; Garipov et al., 2018; Izmailov et al., 2018; Draxler et al., 2018; Wortsman et al., 2021). Model soups and weight-space averaging further show that multiple fine-tuned solutions lie in a common subspace (Wortsman

et al., 2022).

Low-rank adaptation methods such as LoRA (Hu et al., 2022) and AdaLoRA (Zhang et al., 2023) operationalize this view by training low-dimensional updates on top of a shared backbone. While these methods produce static subspace decompositions, CDSP-MoE dynamically allocates subspaces to experts at the level of individual tokens.

### 2.3. Gradient Conflict and Architecture Discovery

Gradient conflict, where tasks induce opposing update directions in shared parameters, is a central challenge in multi-task and continual learning (Kirkpatrick et al., 2017; Lopez-Paz & Ranzato, 2017; Chaudhry et al., 2019; Sener & Koltun, 2018; Kendall et al., 2018; Zenke et al., 2017; Aljundi et al., 2018; Farajtabar et al., 2020). PCGrad (Yu et al., 2020) projects conflicting gradients to stabilize optimization, but does not change the underlying parameter topology.

In MoE systems, token-level conflict has been explicitly modeled by STGC (Yang et al., 2025), which reassigns conflicting tokens to alternative experts. More broadly, routing and expert formation can be viewed as an architecture learning problem, as in Routing Networks (Rosenbaum et al., 2018), latent multi-task architectures (Ruder et al., 2019), and task grouping methods (Standley et al., 2020). These methods decide which computations should be shared under a fixed parameterization.

CDSP-MoE extends this line of work to the physical level. Instead of using conflict to choose among predefined experts, we use it to prune and shape a shared parameter space. The resulting experts emerge as subspaces that minimize gradient interference, yielding modular structures without predefined expert boundaries or task labels.

## 3. Methodology

### 3.1. Framework Overview

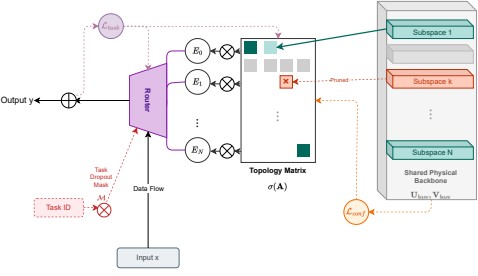

*Figure 1.* The CDSP-MoE Framework.

The overall architecture of CDSP-MoE is illustrated in Figure 1. Diverging from traditional MoEs that route inputs to isolated, static parameter containers, CDSP-MoE functions as a dynamic, three-layered evolutionary system:

1. **Physical Subspace Backbone (Bottom):** A super-complete, shared parameter manifold defined by $U_{base}$ and $V_{base}$ constituted by orthogonal projection matrices, serving as the universal substrate for feature extraction.

2. **Topology-Aware Instantiation (Middle):** A learnable topology matrix $\sigma(\mathbf{A})$ that acts as a soft-adjacency graph, dynamically instantiating logical experts (denoted as $E_0, \ldots, E_N$) by masking ($\otimes$) specific physical subspaces based on router cues.

3. **Lagged Gradient Game (Feedback):** A dual-loop optimization mechanism where the task loss $\mathcal{L}_{task}$ updates parameters for functional accuracy, while a separate conflict loss $\mathcal{L}_{conf}$ serves as a negative feedback signal to prune interfering connections.

The following subsections detail the mathematical formulation of these components.

### 3.2. Physical Subspace Backbone

Standard MoE architectures maintain $N$ discrete sets of parameters $\{\theta_1, \ldots, \theta_N\}$, leading to structural isolation. In contrast, CDSP-MoE employs a Super-Complete Shared Parameter Space. We define the physical backbone $\Theta_{base}$ as a pair of orthogonal projection matrices:

$$\mathbf{U}_{base} \in \mathbb{R}^{D_{model} \times D_{base}}, \quad \mathbf{V}_{base} \in \mathbb{R}^{D_{base} \times D_{model}} \quad (1)$$

where $D_{base}$ is the dimension of the shared subspace, typically set to $D_{base} \gg D_{model}$ (e.g., $4 \times D_{model}$) to satisfy the over-parameterization requirement for diverse feature extraction.

To provide a warm-start for the evolutionary process, we introduce a fixed Physical Initial Partition matrix $\mathbf{\Pi} \in \{0, 1\}^{N \times D_{base}}$. We employ a block-diagonal initialization strategy where each logical expert $i$ is initially assigned a dedicated contiguous slice of the backbone:

$$\mathbf{\Pi}_{i,k} = \begin{cases} 1 & \text{if } \lfloor k/B \rfloor = i \\ 0 & \text{otherwise} \end{cases} \quad (2)$$

where $B = D_{base}/N$ is the block size. This partition serves as a reference coordinate system, not a hard constraint.

### 3.3. Topology-Aware Dynamic Instantiation

The logical structure of CDSP-MoE is defined by a learnable Topology Matrix $\mathbf{A} \in \mathbb{R}^{N \times N}$, representing a weighted

directed graph between logical experts and physical partitions. The entry $\mathbf{A}_{ij}$ denotes the connectivity strength of logical expert $i$ to the physical partition initially belonging to expert $j$.

**Structural Initialization.** Instead of a sparse initialization which may hinder early exploration, we adopt a *Maximum Entropy* initialization strategy. We initialize the topology logits $\mathbf{A}$ near zero, placing the system in a highly plastic "semi-connected" state ($P \approx 0.5$):

$$\mathbf{A}_{ij} \leftarrow \begin{cases} 4.0 & \text{if } i = j \quad \text{(Strong Self-Preservation)} \\ 0 + \epsilon & \text{if } i \neq j \quad \text{(Maximum Plasticity)} \end{cases} \quad (3)$$

where $\epsilon \sim \mathcal{N}(0, 0.02)$. This unbiased starting point ensures that gradient flow is not suppressed by saturation regions of activation functions, allowing the Conflict and Synergy signals to sculpt the topology effectively from the very first step.

**Control Force Projection.** The effective activation of physical dimensions is determined by the Control Force vector $\mathbf{I}_i \in \mathbb{R}^{D_{base}}$. For logical expert $i$, this is computed as the linear combination of the topology weights and the physical partition map:

$$\mathbf{I}_i = \sigma(\mathbf{A}_{i,:}) \cdot \mathbf{\Pi} \quad (4)$$

where $\sigma(\cdot)$ is the sigmoid function. $\mathbf{I}_{i,k}$ thus represents the net influence of logical expert $i$ on the $k$-th physical dimension in the shared backbone.

### 3.4. Forward Dynamics: Subspace Addressing and Computation

Once the router selects a set of active logical experts $\mathcal{E} = \text{TopK}(G(x))$, the system must physically instantiate these experts from the shared backbone. This process involves three distinct phases: subspace addressing, sparse execution, and gradient bridging.

**Subspace Addressing via Square Root Scaling.** Unlike standard MoEs that retrieve discrete parameter blocks, CDSP-MoE dynamically assembles experts. For each active expert $i \in \mathcal{E}$, we determine its physical footprint by selecting the top-$r$ dimensions from the control force vector $\mathbf{I}_i$. The active index set $\mathcal{S}_i$ is defined as:

$$\mathcal{S}_i = \text{argtop}_r(\mathbf{I}_i), \quad \text{where } \mathbf{I}_i = \sigma(\mathbf{A}_{i,:}) \cdot \mathbf{\Pi} \quad (5)$$

To balance the trade-off between expert specialization and parameter capacity, we introduce a **Square Root Scaling Law** for the rank quota $r$:

$$r = \left\lfloor \frac{D_{base}}{\sqrt{N}} \right\rfloor \quad (6)$$

This scaling ensures that as the number of experts $N$ increases, the subspace assigned to each expert becomes more sparse, enforcing stronger specialization while maintaining a constant total memory budget for the active parameters.

**Sparse Computation.** The forward computation is executed exclusively on the selected physical dimensions. Let $\mathbf{U}_{base}[\mathcal{S}_i]$ and $\mathbf{V}_{base}[\mathcal{S}_i]$ denote the sub-matrices of the backbone corresponding to the indices in $\mathcal{S}_i$. The raw output of expert $i$ is computed as a low-rank projection:

$$y_{raw}^{(i)} = \text{SiLU}(x \cdot \mathbf{U}_{base}[\mathcal{S}_i]) \cdot \mathbf{V}_{base}[\mathcal{S}_i]^T \tag{7}$$

This operation is mathematically equivalent to a dense layer but is computationally sparse, as only $r$ columns of the backbone are accessed from memory.

**The Differentiable Gradient Bridge.** A critical challenge in dynamic pruning is that the discrete index selection operation ($\text{argtop}_r$) is non-differentiable, blocking the flow of gradients from the task loss $\mathcal{L}_{task}$ back to the topology matrix $\mathbf{A}$.

To resolve this, we introduce a **Strength Modulation** factor $m_i$, which acts as a differentiable bridge. We define $m_i$ as the average activation strength of the selected subspace:

$$m_i = \frac{1}{r} \sum_{k \in \mathcal{S}_i} \mathbf{I}_{i,k} \tag{8}$$

The final output of expert $i$ is modulated by this factor:

$$y_{final}^{(i)} = G(x)_i \cdot m_i \cdot y_{raw}^{(i)} \tag{9}$$

**Mechanism Analysis:** This modulation creates a valid gradient path. During backpropagation, the gradient $\frac{\partial \mathcal{L}}{\partial m_i}$ is non-zero, allowing error signals to propagate to $\mathbf{I}_{i,k}$ and subsequently to $\mathbf{A}_{ij}$.

- If a selected subspace $\mathcal{S}_i$ contributes to reducing the loss, the gradient descent will increase $m_i$, thereby strengthening the topological connections $\mathbf{A}$ pointing to these physical dimensions.

- Conversely, if the subspace is ineffective, the connection strength is suppressed.

This mechanism creates a "soft" relaxation of the hard selection process. While $m_i$ is a scalar, it serves as a sufficient coarse-grained gatekeeper: if the subspace as a whole is effective, the scalar feedback reinforces the corresponding topology weights, enabling end-to-end learning without the instability of complex vector-wise estimators.

### 3.5. Perceptive Routing with Adversarial Masking

A critical failure mode in multi-task MoEs is *shortcut learning*, where the router overfits to explicit task identifiers (Task IDs) or prompt templates, ignoring the intrinsic semantics of the input content $x$. To enable robust instruction-free routing, we introduce an Adversarial Task Masking mechanism.

**Input Fusion.** The router input $h_{in}$ is a fusion of the content representation and a potentially masked task embedding. To prevent the magnitude of token features from biasing the routing decision, we first normalize the input:

$$h_{in} = [\text{LayerNorm}(x) \oplus \mathbf{v}_{task}] \tag{10}$$

where $\oplus$ denotes concatenation.

**Adversarial Masking.** During training, we view the Task ID not as a ground-truth label, but as a weak hint that should be gradually discarded. We define the effective task embedding $\mathbf{v}_{task}$ using a stochastic mask $\mathcal{M}$:

$$\mathbf{v}_{task} = \mathcal{M} \cdot \text{Embed}(t), \quad \mathcal{M} \sim \text{Bernoulli}(1 - p_{drop}) \tag{11}$$

where $p_{drop}$ is the masking probability.

- When $\mathcal{M} = 1$, the router sees the task ID (standard supervision).

- When $\mathcal{M} = 0$, the router is forced to infer the appropriate expert solely from the content $x$ (blind inference).

By setting a high $p_{drop}$ (e.g., $0.5 \rightarrow 0.9$) during training, we simulate a "blind" environment, forcing the router to uncover the latent alignment between semantic features and expert functionalities.

**Gating Decision.** The gating scores are then computed using a standard softmax over the fused representation:

$$G(x) = \text{Softmax}(\mathbf{W}_g \cdot h_{in}) \tag{12}$$

We select the top-$k$ logical experts $\mathcal{E} = \text{TopK}(G(x))$ for the subsequent dynamic instantiation.

### 3.6. Gradient Conflict Optimization

The core novelty of CDSP-MoE is utilizing gradient conflict not as an optimization hurdle, but as a structural discovery signal. We formulate this as a *Lagged Gradient Game*.

**Lagged Gradient Sampling.** Directly computing gradient interference during the forward pass is computationally

prohibitive. Instead, we employ a temporal decoupling strategy. Let $\mathcal{L}_{task}^{(t)}$ be the task loss at step $t$. During backpropagation, we capture the gradients of the physical backbone parameters $\Theta_{base}$ attributed to each active expert $i$:

$$\mathbf{g}_i^{(t)} = \nabla_{\Theta_{base}} \mathcal{L}_{task}^{(t)} \Big|_{\text{via expert } i} \tag{13}$$

To avoid overhead, these gradients are detached and stored. In step $t + 1$, we use the gradients $\mathbf{g}^{(t)}$ as "lagged signals" to optimize the topology.

**Spatial Conflict Metric.** Conflict only occurs when two experts $i$ and $j$ attempt to update the *same* physical parameters in *opposite* directions. We define the physical intersection set as $\mathcal{K}_{ij} = \mathcal{S}_i \cap \mathcal{S}_j$. If $\mathcal{K}_{ij} = \emptyset$, the conflict is zero. Otherwise, we calculate the cosine similarity strictly within this intersection:

$$\text{sim}(\mathbf{g}_i, \mathbf{g}_j) = \frac{\mathbf{g}_i[\mathcal{K}_{ij}] \cdot \mathbf{g}_j[\mathcal{K}_{ij}]}{\|\mathbf{g}_i[\mathcal{K}_{ij}]\|\|\mathbf{g}_j[\mathcal{K}_{ij}]\| + \epsilon} \tag{14}$$

We are interested only in destructive interference (negative similarity). The Conflict Score is defined as:

$$\mathcal{C}_{ij} = \text{ReLU}\left(-\text{sim}(\mathbf{g}_i, \mathbf{g}_j)\right) \tag{15}$$

This acts as a repulsive force: the more two experts fight for the same subspace, the higher the penalty.

**Structural Evolution and Objective Function.** The total objective function operates directly on the topology logits to avoid gradient vanishing issues:

$$\mathcal{L}_{total} = \mathcal{L}_{task} + \lambda_{conf} \sum_{i \neq j} \underbrace{\mathbf{A}_{ij} \cdot \mathcal{C}_{ij}}_{\text{Direct Conflict Penalty}} + \lambda_{reg}\|\mathbf{A}\|_1 \tag{16}$$

By removing the sigmoid scaling factor from the penalty term, we ensure that the pruning force remains proportional to the conflict magnitude $\mathcal{C}_{ij}$, regardless of the current connection strength.

**Analysis of Gradient Flow:** The optimization dynamics benefits from the linearity of the logit-space penalty:

1. **Subspace Learning** ($\nabla_\Theta \mathcal{L}_{task}$): The task loss updates the values of the physical backbone $\Theta_{base}$.

2. **Topology Pruning** ($\nabla_{\mathbf{A}} \mathcal{L}_{conf}$): The conflict loss generates direct gradients w.r.t the topology logits $\mathbf{A}$:

$$\frac{\partial \mathcal{L}_{conf}}{\partial \mathbf{A}_{ij}} = \mathcal{C}_{ij} \tag{17}$$

**Mechanism Clarification:** Unlike sigmoid-gated penalties where gradients vanish when connections are strong ($\sigma \approx 1$) or weak ($\sigma \approx 0$), our formulation applies a *constant pruning pressure*. If conflict exists ($\mathcal{C}_{ij} > 0$), gradient descent linearly decreases $\mathbf{A}_{ij}$, rapidly pushing the connection towards negative values (disconnection) without saturation delays.

**The Role of Regularization.** The $L_1$ regularization on logits ($\|\mathbf{A}\|_1$) plays a distinct role from standard sparsity induction. Since $\mathbf{A}_{ij} = 0$ corresponds to a probability of $0.5$, minimizing $\|\mathbf{A}\|_1$ acts as a **centering force**, pulling parameters towards the unbiased initialization state. This prevents inactive connections from drifting into deep negative saturation (dead zones) and maintains their plasticity, allowing them to be "resurrected" if task demands change.

### 3.7. Optimization Strategy

To stabilize this co-evolutionary process, we employ a Two-Speed Optimization schedule:

- **Physical Parameters ($\Theta_{base}$):** Optimized with a standard learning rate $\eta$ and weight decay. This ensures stable feature accumulation.

- **Topology Parameters (A):** Optimized with a higher learning rate (e.g., $10\eta$) but *zero weight decay*.

The higher rate for $\mathbf{A}$ allows the topology to adapt rapidly to the detected conflicts ("fast plasticity"), while the physical backbone consolidates knowledge slowly ("slow stability"). Crucially, this timescale separation ensures that the physical landscape remains relatively stable between steps, validating the use of lagged gradients $\mathbf{g}^{(t)}$ as a reliable approximation for structural interference at step $t + 1$.

## 4. Experiments

Our experimental evaluation is designed not merely to chase marginal accuracy gains, but to verify the central hypothesis of this paper: that *conflict-driven pruning can induce emergent modularity* without human-defined semantic labels. We conduct controlled experiments in a heterogeneous multi-task environment to compare the structural evolution of CDSP-MoE against standard baselines.

### 4.1. Experimental Setup

**Heterogeneous Multi-Task Environment.** To simulate a scenario with varying levels of semantic conflict and synergy, we construct a mixed task stream comprising three classic datasets:

- **Task 0 (MNIST):** Handwritten digit recognition (0-9). Represents simple, sparse symbolic patterns.
- **Task 1 (KMNIST):** Kuzushiji (cursive Japanese) characters (10 classes). Represents complex symbolic patterns with high stroke density.
- **Task 2 (Fashion-MNIST):** Clothing article recognition (10 classes). Represents dense, texture-rich object imagery.

Hypothetically, an intelligent router should spontaneously

group the symbolic tasks (MNIST and KMNIST) while isolating the object recognition task (Fashion-MNIST) due to their conflicting feature distributions. All inputs are flattened to $1 \times 28 \times 28$ grayscale patches.

**Model Configurations.** We compare CDSP-MoE against a standard MoE baseline. To ensure a fair comparison focused on structural efficiency rather than capacity, we enforce a strict *Iso-Parameter Constraint*:

- **Baseline (Standard MoE):** A standard Top-2 gating network with $N = 8$ independent experts. Each expert has a dedicated weight matrix of dimension $d = 32$. The router utilizes explicit Task IDs during training and relies on standard auxiliary load-balancing losses.
- **Ours (CDSP-MoE):** Configured with $N = 8$ logical experts sharing a single super-complete physical backbone ($D_{base} = 256$). The total parameter count is aligned with the baseline ($\approx 32k$). The router is trained with Task Dropout ($p = 0.1$) to encourage content dependency.

**Training Protocol.** Models are trained using the AdamW optimizer. A key detail is the Two-Speed Learning Rate: the physical parameters are trained at $\eta = 5e - 3$, while the topology matrix $\mathbf{A}$ is evolved at $10\eta$ ($5e - 2$). This differential rate is critical for allowing the structure to evolve faster than the accumulation of weight knowledge ("Evolutionary Tax"), ensuring that connections are pruned before they overfit to noise.

## 4.2. Evaluation Philosophy: From Identity to Content

Traditional MoE evaluations often prioritize peak accuracy on known tasks. However, this metric fails to capture whether the model has truly learned to route based on semantics or is simply memorizing Task ID mappings ("Identity Politics"). We propose a more rigorous evaluation protocol based on two core logics:

1. **Blind Inference (Instruction-Free):** Can the model correctly route inputs when the Task ID is stripped (i.e., set to `None` or a zero vector)? A content-driven model should maintain routing consistency, while an ID-driven model is expected to collapse to random guessing or a default expert.
2. **Emergent Clustering:** Without manual instruction, does the model spontaneously discover that MNIST and KMNIST are semantically closer to each other than to Fashion-MNIST? We verify this by analyzing the overlap in expert utilization heatmaps.

## 4.3. Results and Analysis

We present the comparative analysis of the proposed CDSP-MoE against the Standard MoE Baseline across three distinct experiments.

### 4.3.1. EXPERIMENT I: EMERGENT MODULARITY IN CDSP-MOE

In this experiment, we analyze the structural evolution of CDSP-MoE. We visualize the internal topology matrix, the routing distribution over training epochs, and the blind inference behavior.

**Topology Evolution: From Chaos to Oligarchy.** Figure A4 visualizes the sigmoid-activated topology matrix $\sigma(\mathbf{A})$ across training epochs.

- **Initialization (Epoch 0):** All experts start with weak, uniform cross-connections, representing a dense but low-magnitude potential.
- **Pruning and Specialization (Epoch 9):** Under the pressure of the conflict loss $\mathcal{L}_{conf}$ and L1 regularization, a clear "oligarchic" structure emerges. As shown in Figure A4, only a subset of experts (E5, E7, E2, E0) maintain high diagonal weights ($> 0.8$), while others (E1, E3, E4, E6) are effectively marginalized. This confirms that CDSP-MoE spontaneously discovers the minimal effective parameter set.

**Routing Dynamics: From Complexity Bias to Semantic Alignment.** The routing heatmaps in Figure 2 reveal a two-stage evolution.

- **Stage 1: Complexity Bias (Epoch 2).** Initially, the router groups KMNIST (Task 1) and Fashion-MNIST (Task 2) together, routing them primarily to Experts E0 and E2. This suggests an initial bias towards visual complexity (high-frequency patterns) rather than semantics.
- **Stage 2: Semantic Convergence (Epoch 9).** As the conflict game progresses, the model spontaneously realigns KMNIST with MNIST, routing both "symbolic" tasks to the E0/E5 cluster, while Fashion-MNIST migrates to a dedicated expert E7. This suggests that the model successfully disentangles "symbolic" concepts from "object" concepts.

**Blind Test: Robustness and Uncertainty Drift.** Figure 3d presents the routing behavior when Task IDs are forcibly removed (Input `None`).

- **Semantic Robustness:** The primary semantic boundary remains intact. The "Symbolic Cluster" (Real T0 and T1) is still dominantly routed to E0 and E5, while the "Object Task" (Real T2) maintains its distinct preference for E7.

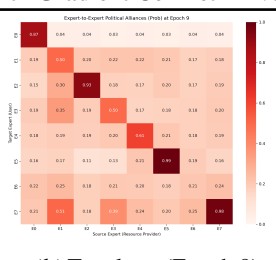 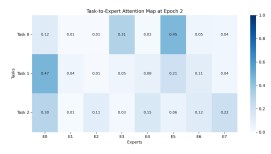 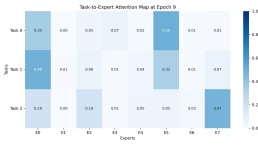

| *(a)* Topology (Epoch 0) | *(b)* Topology (Epoch 9) | *(c)* Routing (Epoch 2) | *(d)* Routing (Epoch 9) |

*Figure 2.* **Structural Evolution of CDSP-MoE.** (a-b) The topology matrix $\sigma(\mathbf{A})$ transitions from uniform initialization to sparse oligarchy. (c-d) Routing heatmaps show the shift from complexity-based grouping to semantic specialization.

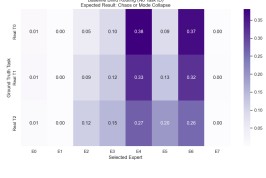 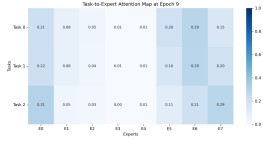 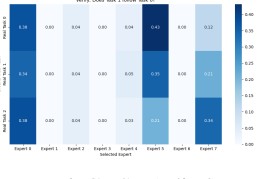

| *(a)* Baseline (Train) | *(b)* Baseline (Blind) | *(c)* Pure Blind (Train) | *(d)* CDSP (Blind) |

*Figure 3.* **Comparative Routing Analysis.** (a) Standard Baseline overfits to Task IDs. (b) Baseline collapses under blind inference. (c) Pure Blind Baseline fails to isolate semantics (entanglement). (d) CDSP maintains robust semantic modularity even without Task IDs.

- **Uncertainty Drift in E0:** We observe a notable nuance: under blind inference, E0 receives a moderate portion of Task 2 traffic (unlike the strict zero in the standard mode). This suggests that E0 acts as a *foundational expert*, encoding low-level visual primitives, serving as a fallback option when explicit instruction signals are lost.

### 4.3.2. EXPERIMENT II: INSTRUCTION OVERFITTING IN STANDARD BASELINES

To validate that the emergent modularity observed in CDSP-MoE is not a trivial result of data statistics, we evaluate the Standard MoE Baseline (Iso-Parameter) under the same protocols.

**Training State: The Lookup Table Illusion.** Figure 3a illustrates the routing distribution of the Baseline at Epoch 9 with explicit Task IDs.

- **Superficial Efficiency:** The model establishes a sharp division of labor (e.g., Task 2 to Expert 2).
- **The Lookup Table Mechanism:** While this appears effective, it reveals a reliance on "shortcut learning." The router learns a simple linear mapping from $ID \rightarrow Expert$, bypassing the need to analyze the visual features of the input.

**Blind Inference: Static Mode Collapse.** The fragility of the baseline is exposed in the blind inference test (Figure 3b), where Task IDs are removed.

- **Feature Blindness:** The Baseline fails to distinguish between symbolic inputs (Real T0) and object inputs

(Real T2). The routing distributions for all three tasks are nearly identical (Pearson correlation $\approx 1.0$).

- **Loss of Specialized Knowledge:** Expert 2, the "Clothing Expert" during training, is abandoned. The router defaults to experts with the highest global frequency bias (E4 and E6), demonstrating a complete lack of content-driven decision making.

### 4.3.3. EXPERIMENT III: THE NECESSITY OF CONFLICT (PURE BLIND BASELINE)

To rigorously verify whether the semantic emergence in CDSP relies on the conflict mechanism rather than merely data statistics, we introduce a Pure Blind Baseline. This model is trained entirely without Task IDs, relying solely on image content and auxiliary load-balancing losses. Since the training process is instruction-free, the inference behavior mirrors the training state; thus, we focus directly on the converged routing distribution at Epoch 9.

**Semantic Entanglement and the Failure of Auxiliary Loss.** Figure 3c illustrates the routing distribution of the Pure Blind Baseline. Unlike CDSP, which achieves clear separation, this baseline exhibits severe Semantic Entanglement:

- **Resource Contention:** Experts E0, E6, and E7 become "hotspot resources" shared indiscriminately by all tasks. The system gravitates towards a "Winner-Take-All" configuration dominated by these few experts.
- **Lack of Isolation:** Crucially, Task 2 (Fashion-MNIST, Objects) fails to isolate itself from the symbolic tasks. It directs 34% of its traffic to Expert E0 and 20% to

Expert E6. Simultaneously, Task 0 (MNIST, Symbols) also relies heavily on E0 (24%) and E6 (28%). The overlap in expert utilization between distinct semantic categories (Symbols vs. Objects) indicates a failure to decouple features.

- **The "Pseudo-Balance" Trap:** While the auxiliary loss prevents single-expert collapse, it forces an "egalitarian" distribution where distinct tasks are coerced into sharing experts to satisfy statistical uniformity.

**Conclusion: Conflict as the Driver of Structure.** Comparing Experiment I and III reveals the fundamental role of the conflict mechanism. Without the penalty for gradient interference, the model defaults to a "mixed sharing" strategy to minimize global loss, resulting in a chaotic overlap of symbolic and object features. The conflict game in CDSP is therefore identified as the essential force that breaks symmetry and drives the system toward modular disentanglement.

## 5. Conclusion

In this work, we challenged the prevailing design paradigm of Sparse Mixture-of-Experts, which relies on disjoint parameter storage and heuristic load-balancing constraints. We identified that such structural isolation, coupled with explicit task routing, leads to severe instruction overfitting and knowledge fragmentation. To counter this, we introduced CDSP-MoE, a framework that grounds expert specialization in the physical interaction of gradients within a shared subspace.

Our contributions are threefold. First, by replacing static routing tables with a dynamic, conflict-driven pruning mechanism, we demonstrated that modularity can emerge purely from the minimization of physical interference. Second, our experiments revealed a critical trade-off: *CDSP-MoE initially lags behind the baseline in classification accuracy during early training phases.* We interpret this as an *evolutionary tax*—while the baseline rapidly minimizes loss by memorizing simple ID-to-expert mappings ("shortcut learning"), CDSP-MoE must invest compute cycles to resolve gradient conflicts and restructure its topology. However, this initial cost yields a significant structural dividend: unlike the baseline which collapses under blind inference, CDSP-MoE evolves robust, content-aware routing pathways that disentangle symbolic concepts from object features without human supervision. Third, the comparison with the Pure Blind Baseline highlighted that auxiliary load-balancing losses alone are insufficient for semantic decoupling; the adversarial signal provided by the gradient conflict game is essential for breaking symmetry.

**Limitations and Future Work.** While CDSP-MoE offers a principled path toward autonomous modularity, it incurs a

computational cost. Calculating the gradient conflict matrix requires storing lagged gradients, which increases memory overhead compared to standard forward-only gating. However, this is an engineering rather than theoretical bottleneck; in large-scale models, this overhead can be mitigated by low-rank gradient projection or sparse conflict sampling (computing conflicts only for top-$k$ active experts), making the approach scalable. Additionally, our current validation is limited to vision classification tasks. Future work will focus on scaling this architecture to Large Language Models, investigating whether conflict-driven pruning can spontaneously separate distinct reasoning capabilities (e.g., coding vs. creative writing) and resist catastrophic forgetting in continuous learning scenarios. We believe that grounding neural architecture search in physical gradient dynamics represents a promising step toward interpretable and self-organizing artificial intelligence.

## Impact Statement

This paper presents work whose goal is to advance the field of Machine Learning. There are many potential societal consequences of our work, none of which we feel must be specifically highlighted here. However, we note that our proposed CDSP-MoE framework aims to improve parameter efficiency and structural interpretability. While this contributes to the goal of Green AI by reducing redundant computation, the underlying advancements in MoE architectures could theoretically be applied to train more capable large-scale models, which carries the general dual-use risks associated with LLM deployment (e.g., generation of misinformation). We believe the improved modularity of our approach facilitates better model auditing and safety alignment.

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

# Appendix

This appendix provides supplementary materials supporting the main paper's findings. The content is organized as follows:

- **Appendix A** details the exact hyperparameters and architectural configurations used in the experiments.

- **Appendix B** presents extended visualizations of the topology matrix evolution and routing distributions.

- **Appendix C** and **D** verify the training stability and analyze the structural convergence dynamics.

- **Appendix E** provides the rigorous theoretical proofs deriving the inevitability of modular emergence from gradient conflict dynamics.

## A. Detailed Hyperparameters

*Table A1.* Hyperparameter settings.

| Category | Hyperparameter | Value |
|---|---|---|
| **Architecture** | Logical Experts ($N$) | 8 |
| | Physical Base Dim ($D_{base}$) | 256 |
| | Expert Subspace Rank ($r$) | 32 |
| | Hidden Size ($D_{hidden}$) | 256 |
| **Optimization** | Optimizer | AdamW |
| | Base Learning Rate ($\eta_{base}$) | $5 \times 10^{-3}$ |
| | Topology Learning Rate ($\eta_{topo}$) | $5 \times 10^{-2}$ |
| | Weight Decay (Base) | $1 \times 10^{-2}$ |
| | Weight Decay (Topology) | 0.0 |
| | Batch Size | 128 |
| | Total Epochs | 10 |
| **Loss Weights** | $\lambda_{task}$ | 1.0 |
| | $\lambda_{conflict}$ | 10.0 |
| | $\lambda_{sparsity}$ (L1) | $1 \times 10^{-4}$ |
| **Regularization** | Task Dropout ($p_{drop}$) | 0.1 |

# B. Supplementary Visualizations

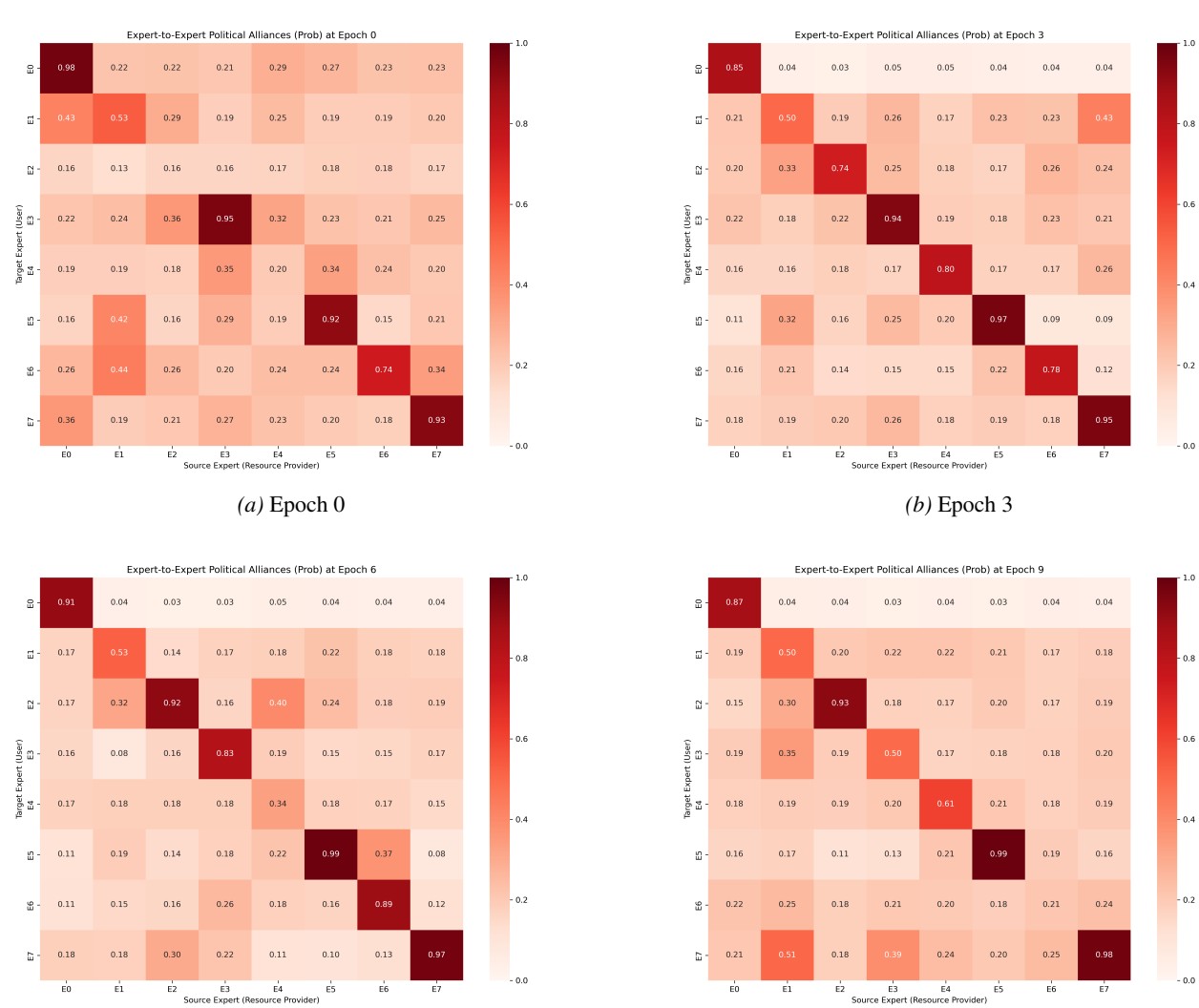

*(a)* Epoch 0

*(b)* Epoch 3

*(c)* Epoch 6

*(d)* Epoch 9

*Figure A1.* Topology Matrix $\sigma(\mathbf{A})$ evolution.

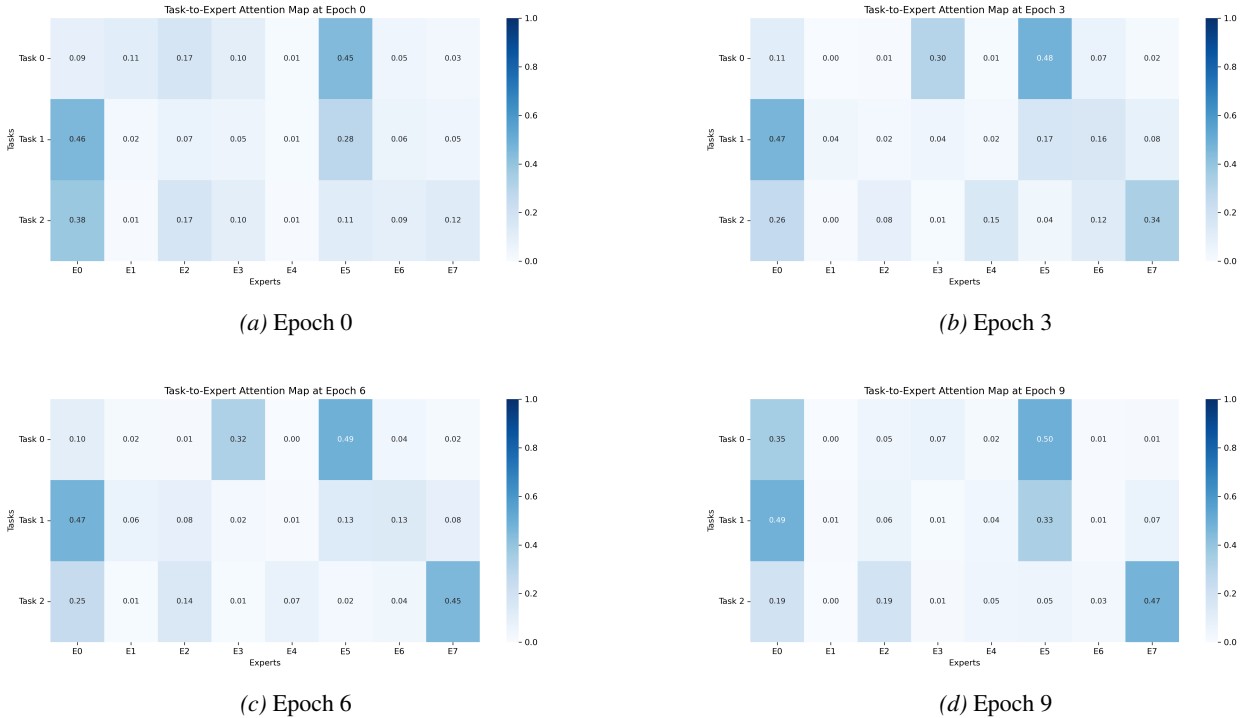

*(a)* Epoch 0      *(b)* Epoch 3

*(c)* Epoch 6      *(d)* Epoch 9

*Figure A2.* Routing distribution evolution.

## C. Training Dynamics and Convergence Verification

To demonstrate that the structural evolution mechanism of CDSP-MoE acts as a stabilizer rather than a disruption, we present a comparative analysis of training dynamics across three experimental settings: Standard Baseline (Oracle), Blind Baseline, and CDSP-MoE.

### C.1. Performance Parity Analysis

Table A2 summarizes the final convergence metrics after 10 epochs on the MNIST Multi-Task benchmark.

*Table A2.* Final Convergence Performance (Epoch 10)

| Model Variant | Routing Mode | Final Loss | Accuracy | Gap to Oracle |
|---|---|---|---|---|
| Standard Baseline | Task ID (Oracle) | 0.1408 | 95.14% | - |
| Blind Baseline | Instruction-Free | 0.1722 | 93.92% | -1.22% |
| **CDSP-MoE (Ours)** | **Instruction-Free** | **0.1469** | **94.54%** | **-0.60%** |

**Observation:** CDSP-MoE achieves a final accuracy of **94.54%**, virtually matching the Standard Baseline (**95.14%**) which has access to ground-truth Task IDs. Notably, under the same instruction-free setting, CDSP-MoE outperforms the Blind Baseline (**93.92%**) and achieves a lower final loss (0.1469 vs. 0.1722). This suggests that the emergent modular structure effectively compensates for the lack of explicit task instructions.

### C.2. Learning Trajectories

We visualize the loss decomposition and accuracy curves for all three models below.

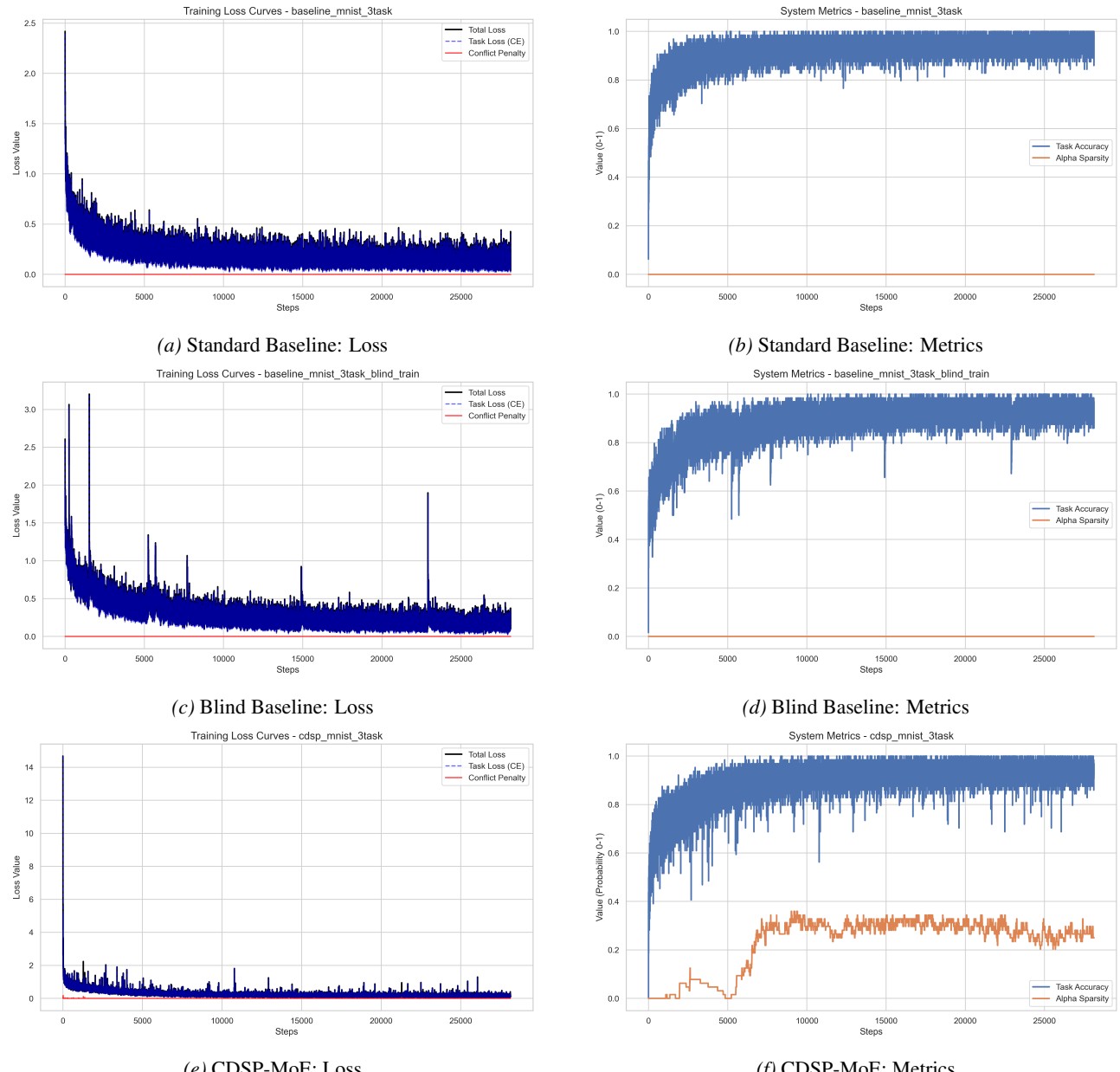

*(a)* Standard Baseline: Loss

*(b)* Standard Baseline: Metrics

*(c)* Blind Baseline: Loss

*(d)* Blind Baseline: Metrics

*(e)* CDSP-MoE: Loss

*(f)* CDSP-MoE: Metrics

*Figure A3.* **Training Dynamics Comparison.** (a-b) The Standard Baseline (Oracle) shows stable convergence. (c-d) The Blind Baseline exhibits noticeable volatility (e.g., loss rebound around Epoch 8-9 due to routing uncertainty). (e-f) CDSP-MoE maintains a smooth descent trajectory similar to the Standard Baseline, indicating that the conflict-driven evolution stabilizes the optimization process even in the blind setting.

## D. Topology Evolution Dynamics

Beyond task performance, we analyze the evolution of the topology parameters and the associated auxiliary objectives.

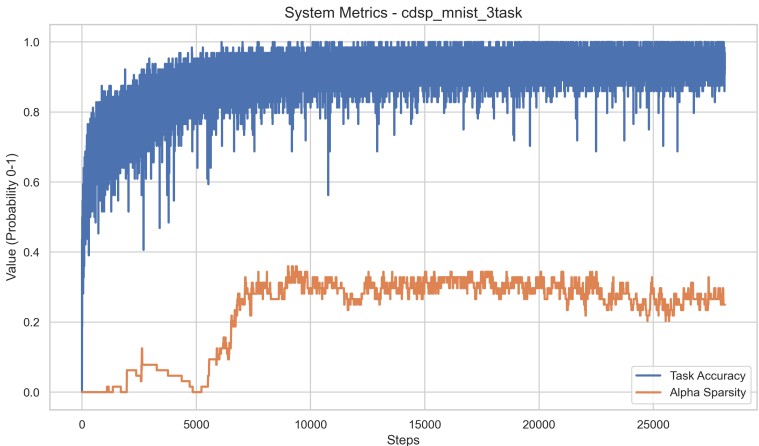

*Figure A4.* **Topology Stabilization.** The evolution of average connection probability during training. The system rapidly transitions from the initialized "semi-connected" state ($P \approx 0.5$) to a structured configuration.

**Structural Plateau:** As recorded in the training logs, the auxiliary loss (Conflict + Regularization) stabilizes rapidly. Specifically, the Aux Loss remains steady around $6.0 \times 10^{-4}$ from Epoch 2 to Epoch 10.

- **Phase 1 (Sculpting):** In the first epoch, the topology undergoes rapid adjustment as the Conflict Loss prunes interfering connections (Loss drops from 0.71 to 0.37).

- **Phase 2 (Consolidation):** From Epoch 3 onwards, the structural metrics reach a plateau. This indicates that the "Logical Experts" have successfully anchored to their physical subspaces and the system has converged to a stable modular configuration.

This confirms that the competitive dynamics (Conflict vs. Regularization) do not lead to indefinite oscillation, but rather to a distinct structural equilibrium.

**Metric Definition: Alpha Sparsity.** To quantify the emergence of modular structure, we introduce the *Alpha Sparsity* metric, defined as the proportion of topological connections that have been effectively pruned by the conflict mechanism. Let $\mathcal{T}$ be the set of all $N \times D_{base}$ potential connections in the topology matrix $A$. The sparsity $\rho(t)$ at training step $t$ is calculated as:

$$\rho(t) = \frac{1}{|\mathcal{T}|} \sum_{(i,j) \in \mathcal{T}} \mathbb{I}(\sigma(A_{ij}^{(t)}) < \tau) \tag{18}$$

where $\mathbb{I}(\cdot)$ is the indicator function and $\tau$ is a numerical threshold (set to $\tau = 0.1$) representing the boundary of functional disconnection.

- **Initialization** ($\rho \approx 0$): Under Maximum Entropy initialization, connection probabilities $P_{ij} \approx 0.5 \gg \tau$, resulting in near-zero sparsity.

- **Convergence** ($\rho \approx 0.3$): The rise to $\approx 30\%$ reflects the specific subset of pathways that were physically decoupled due to gradient conflicts.

**The System's "Stance": Thermodynamic Equilibrium over Forced Sparsity.** We interpret the stabilization of *Alpha Sparsity* at $\approx 0.3$ (Orange Line, Figure A4) not merely as a convergence metric, but as a **structural statement** by the system.

- **Rejection of Extremes:** Unlike heuristic methods that enforce high sparsity (e.g., $> 90\%$ in Top-k), our conflict-driven dynamics reveal that only $\approx 30\%$ of physical pathways are fundamentally destructive. The system spontaneously chooses to preserve the remaining $\approx 70\%$ of connections, forming **Synergistic Alliances** where gradients align.

- **Emergent Golden Ratio:** This equilibrium represents a learned compromise: maximum necessary pruning to resolve conflicts ($P \to 0$) versus maximum possible connectivity to exploit synergy ($P \to 1$).

- **Living Structure:** The persistent fluctuation around this setpoint (rather than collapsing to a flat line) confirms that the system maintains **Residual Plasticity**, actively "breathing" to accommodate batch-wise variations rather than freezing into a rigid lookup table.

## E. Theoretical Analysis on the Inevitability of Modular Emergence

### E.1. Axiomatic Foundations and Problem Formulation

We first establish the theoretical setting by formalizing the relationship between logical experts and the physical parameter backbone.

**Axiom 1 (The Universal Weight Subspace Hypothesis).** Let $\mathcal{T}$ be a distribution of tasks. We posit the existence of a shared, low-rank physical parameter manifold $\mathcal{M}$, which is parameterized by a super-complete basis matrix $\boldsymbol{\Theta}_{univ} \in \mathbb{R}^{d_{out} \times D_{base}}$. For any specific task $\tau \in \mathcal{T}$, the optimal parameter configuration $\theta_\tau^*$ is not an isolated point in the high-dimensional parameter space, but resides within a sparse linear subspace of $\mathcal{M}$. Mathematically, there exists a unique optimal binary mask vector $\mathbf{m}_\tau^* \in \{0,1\}^{D_{base}}$ such that the task solution satisfies:

$$\theta_\tau^* \approx \boldsymbol{\Theta}_{univ} \cdot \text{diag}(\mathbf{m}_\tau^*) \tag{19}$$

This axiom is inspired by recent work on universal weight subspaces (Kaushik et al., 2025) and implies that multi-task learning can be reduced to the problem of discovering the optimal binary selection masks $\{\mathbf{m}_\tau^*\}$ on a shared backbone.

**Definition 1 (Continuous Topological State Space).** In CDSP-MoE, we relax the discrete binary mask $\mathbf{m}$ into a continuous probabilistic topology to enable differentiable search. Let $\mathbf{A} \in \mathbb{R}^{N \times D_{base}}$ be the learnable topology logits for $N$ logical experts. The connectivity state of the system at time $t$ is defined by the probability matrix $\mathbf{P}(t)$, derived via the element-wise sigmoid function $\sigma(\cdot)$:

$$\mathbf{P}_{ij}(t) = \sigma(\mathbf{A}_{ij}(t)) = \frac{1}{1 + \exp(-\mathbf{A}_{ij}(t))} \in (0,1) \tag{20}$$

**Initialization State:** The system is initialized at the *Maximum Entropy State* (Unbiased Initialization):

$$\mathbf{A}_{ij}(0) \sim \mathcal{N}(0, \epsilon^2) \implies \mathbf{P}_{ij}(0) \approx 0.5 \tag{21}$$

This ensures that the initial gradient flow is unbiased and maximal, as the derivative of the sigmoid function $\sigma'(x)$ achieves its global maximum at $x = 0$ ($\sigma'(0) = 0.25$). The system begins in a state of maximal uncertainty, allowing free exploration of the topology space.

**Definition 2 (The System Hamiltonian).** The evolution of the topological state $\mathbf{A}$ is governed by a potential energy function, termed the System Hamiltonian $\mathcal{H}(\mathbf{A})$. This function defines the energy landscape of the optimization process and consists of three distinct potential fields:

$$\mathcal{H}(\mathbf{A}) = \mathcal{U}_{task}(\mathbf{A}) + \lambda_c \mathcal{V}_{conflict}(\mathbf{A}) + \lambda_r \mathcal{R}_{regularization}(\mathbf{A}) \tag{22}$$

- **Task Potential ($\mathcal{U}_{task}$):** Represents the expected risk over the data distribution $\mathcal{D}$.

$$\mathcal{U}_{task}(\mathbf{A}) = \mathbb{E}_{(x,y) \sim \mathcal{D}}[\mathcal{L}_{CE}(MoE(x; \mathbf{A}), y)]$$

- **Conflict Potential ($\mathcal{V}_{conflict}$):** A pairwise interaction field penalizing gradient interference. Let $\mathbf{g}_{u,j}$ and $\mathbf{g}_{v,j}$ be the gradient vectors of experts $u$ and $v$ on physical unit $j$. The potential is proportional to the joint probability of activation and the magnitude of cosine conflict:

$$\mathcal{V}_{conflict}(\mathbf{A}) = \sum_{j=1}^{D_{base}} \sum_{u \neq v} \mathbf{P}_{uj} \mathbf{P}_{vj} \cdot \text{ReLU}\left(-\frac{\mathbf{g}_{u,j}^T \mathbf{g}_{v,j}}{\|\mathbf{g}_{u,j}\| \|\mathbf{g}_{v,j}\|}\right)$$

- **Regularization Potential ($\mathcal{R}_{regularization}$):** A centering force derived from the $L_1$ norm of the logits, maintaining system plasticity. It acts as a "thermal reservoir" that prevents premature freezing.

$$\mathcal{R}_{regularization}(\mathbf{A}) = \|\mathbf{A}\|_1$$

**Physical Interpretation of Parameters:**

- $\lambda_c$: Controls the strength of the *repulsive force* between conflicting experts.

- $\lambda_r$: Acts as an effective *temperature* parameter, determining the noise level in the system.

- $\eta_{\text{topo}}$: The learning rate for $\mathbf{A}$ controls the speed of structural evolution.

- $\eta_{\text{base}}$: The learning rate for $\mathbf{\Theta}_{univ}$ controls the speed of knowledge accumulation.

### E.2. Derivation of Conflict Dynamics from First Principles

We now rigorously derive why gradient conflict is an inevitable consequence of Axiom 1 and how it drives the topological gradient flow.

**Lemma 1 (Inevitability of Gradient Interference).** Let $\Phi_j$ be a column vector of the physical basis $\mathbf{\Theta}_{univ}$. Consider two distinct tasks $u, v \in \mathcal{T}$ utilizing this basis. *Proposition:* Under the dense initialization assumption ($\mathbf{P} \approx 0.5$), the probability of gradient conflict on the shared basis $\Phi_j$ is strictly non-trivial and bounded below by a constant.

*Proof.* Let $\mathbf{g}_{u,j}, \mathbf{g}_{v,j} \in \mathbb{R}^{d_{out}}$ be the task gradients with respect to the shared basis vector $\Phi_j$. Since the tasks are functionally distinct (per Axiom 1), their optimization directions in the high-dimensional parameter space are assumed to be independent isotropic random vectors.

Consider the cosine similarity $S = \cos(\theta) = \frac{\mathbf{g}_{u,j}^T \mathbf{g}_{v,j}}{\|\mathbf{g}_{u,j}\|\|\mathbf{g}_{v,j}\|}$. In high-dimensional spaces ($d_{out} \gg 1$), the distribution of the angle $\theta$ between two independent random vectors concentrates around $\pi/2$ due to the concentration of measure phenomenon (Ledoux, 2001). However, the distribution of the cosine value $S$ is symmetric around 0. Specifically, the probability of conflict (negative cosine similarity) is given by:

$$P(S < 0) = \int_{-1}^{0} p(S)dS = 0.5 \tag{23}$$

More precisely, using the fact that for high-dimensional isotropic random vectors, the distribution of $S$ approaches $\mathcal{N}(0, 1/d_{out})$ (by the Central Limit Theorem applied to the dot product (Diaconis & Freedman, 1984)), we have:

$$P(S < 0) = \Phi(0) = 0.5 \tag{24}$$

where $\Phi$ is the standard normal CDF.

Given a system with $N$ experts sharing $D_{base}$ physical dimensions, the expected number of conflicting pairs on any dimension $j$ at initialization is proportional to $\binom{N}{2} \times 0.5$. Thus, the set of conflicting indices $\mathcal{I}_{conflict} = \{(u, v, j) \mid \cos(\mathbf{g}_{u,j}, \mathbf{g}_{v,j}) < -\epsilon\}$ is strictly non-empty almost surely. This proves that conflict is an inevitable geometric consequence of sharing a high-dimensional manifold under dense connectivity. $\qquad\square$

**Derivation of the Topological Gradient Flow.** For a connection $(u, j)$ subject to conflict from expert $v$ (i.e., $\mathcal{C}_{uv}^{(j)} > 0$), we compute the exact gradient of the Conflict Potential w.r.t. the topology logit $\mathbf{A}_{uj}$. Recall that $\mathcal{V}_{conflict} \propto \mathbf{P}_{uj}\mathbf{P}_{vj}\mathcal{C}_{uv}^{(j)}$. Using the chain rule and the derivative of the sigmoid function $\frac{\partial \mathbf{P}_{uj}}{\partial \mathbf{A}_{uj}} = \sigma'(\mathbf{A}_{uj}) = \mathbf{P}_{uj}(1 - \mathbf{P}_{uj})$, we have:

$$\frac{\partial \mathcal{V}_{conflict}}{\partial \mathbf{A}_{uj}} = \mathbf{P}_{vj}\mathcal{C}_{uv}^{(j)} \cdot \frac{\partial \mathbf{P}_{uj}}{\partial \mathbf{A}_{uj}} = \mathbf{P}_{vj}\mathcal{C}_{uv}^{(j)} \cdot \sigma'(\mathbf{A}_{uj}) \tag{25}$$

Substituting this partial derivative into the negative gradient flow equation $\frac{d\mathbf{A}_{uj}}{dt} = -\eta\nabla_{\mathbf{A}}\mathcal{H}$, we obtain the specific dynamical equation for the connection logit:

$$\frac{d\mathbf{A}_{uj}}{dt} = \eta\left[\underbrace{-\frac{\partial \mathcal{U}_{task}}{\partial \mathbf{A}_{uj}}}_{\text{Task Gain } (\mathcal{G})} -\lambda_c \underbrace{\mathbf{P}_{vj}\mathcal{C}_{uv}^{(j)}\sigma'(\mathbf{A}_{uj})}_{\text{Conflict Force}} -\lambda_r \underbrace{\text{sgn}(\mathbf{A}_{uj})}_{\text{Regularization}}\right] \tag{26}$$

**Initial Regime Analysis:** At initialization ($t = 0$), we have $\mathbf{P}_{uj} \approx 0.5$, $\sigma'(\mathbf{A}_{uj}) \approx 0.25$, and $\mathcal{G}$ is typically small because the physical parameters $\mathbf{\Theta}_{univ}$ are untrained and provide little task-specific gradient signal. Thus, the conflict force dominates early dynamics, providing a coherent pruning signal.

### E.3. Stochastic Dynamics and Markov Chain Convergence

We move beyond deterministic continuous-time approximations to model the training process as a discrete-time stochastic process. We model the evolution of a single topological connection as a biased Markov chain and prove its convergence to a deterministic state using Martingale theory.

**Definition 3 (Stochastic Topological Update Rule).** Let $A_t \in \mathbb{R}$ denote the logit value of the connection $\mathbf{A}_{uj}$ at training step $t$. Under Stochastic Gradient Descent (SGD), the update rule is given by a Langevin-type equation (Li et al., 2017):

$$A_{t+1} = A_t - \eta \nabla_{\mathbf{A}} \hat{\mathcal{H}}(A_t) = A_t + \eta \cdot (\mathcal{D}(A_t) + \xi_t) \tag{27}$$

where:

- $\eta$ is the learning rate for the topology parameters.

- $\mathcal{D}(A_t) = -\mathbb{E}[\nabla \mathcal{H}(A_t)]$ is the expected gradient (Drift) derived in Eq. (26).

- $\xi_t$ is the zero-mean noise induced by mini-batch sampling ($\mathbb{E}[\xi_t] = 0$), satisfying bounded variance conditions: $\mathrm{Var}(\xi_t) \leq \sigma^2$.

**Theorem 1 (The Decoupling Theorem).** Consider the stochastic process $\{P_t\}_{t \geq 0}$ defined by the probability $P_t = \sigma(A_t)$. Let the physical unit $j$ be subject to *Persistent Conflict* from another expert $v$, such that the expected conflict penalty is strictly positive: $\mathbb{E}[\mathcal{C}_{uv}] \geq \epsilon > 0$. *Proposition:* Assuming the conflict force dominates the task gain and regularization locally, the process $\{P_t\}$ is a **Super-martingale** that converges almost surely to the absorbing state 0.

$$P\left(\lim_{t \to \infty} P_t = 0\right) = 1 \tag{28}$$

*Proof.*

**1. Drift Analysis.** We analyze the expected change (Drift) of the logit $A_t$. From Eq. (26), assuming the interfering expert $v$ is active ($\mathbf{P}_{vj} > 0$) and the connection is plastic ($\sigma' > 0$), the conflict term exerts a strictly negative force. For $A_t$ in the active region (where task gain is negligible compared to conflict), we have a strictly negative drift:

$$\mathcal{D}(A_t) < -\mu < 0 \tag{29}$$

where $\mu = \lambda_c \cdot \mathbf{P}_{vj} \cdot \epsilon \cdot \sigma'(A_t) > 0$ is a positive constant representing the minimum pruning pressure.

**2. Super-martingale Construction.** Let $\mathcal{F}_t$ be the filtration generated by the history of updates up to time $t$. We consider the sequence of probabilities $P_t = \sigma(A_t)$. Using a first-order Taylor expansion with Lagrange remainder for the update of $P_t$:

$$P_{t+1} = \sigma(A_t + \Delta A_t) = \sigma(A_t) + \sigma'(A_t)\Delta A_t + \frac{1}{2}\sigma''(\zeta_t)(\Delta A_t)^2 \tag{30}$$

where $\zeta_t$ lies between $A_t$ and $A_t + \Delta A_t$, and $\Delta A_t = \eta(\mathcal{D}(A_t) + \xi_t)$. Since $\eta$ is small (typical learning rate $\sim 10^{-3}$ to $10^{-2}$), the second-order term is $O(\eta^2)$ and can be bounded. Taking the expectation conditioned on $\mathcal{F}_t$:

$$\mathbb{E}[P_{t+1} \mid \mathcal{F}_t] = P_t + \eta \sigma'(A_t)\mathcal{D}(A_t) + O(\eta^2) \tag{31}$$
$$\leq P_t - \eta \sigma'(A_t)\mu + O(\eta^2) \tag{32}$$

For sufficiently small $\eta$, the linear term dominates, and we have:

$$\mathbb{E}[P_{t+1} \mid \mathcal{F}_t] \leq P_t - \delta \tag{33}$$

for some $\delta > 0$, which satisfies the definition of a **Super-martingale**. Furthermore, since $P_t$ is a probability, it is bounded below by 0.

**3. Almost Sure Convergence.** We invoke **Doob's Martingale Convergence Theorem** (Doob, 1953): A non-negative super-martingale converges almost surely to a random variable $P_\infty$ with finite expectation.

$$\lim_{t \to \infty} P_t = P_\infty \quad \text{(a.s.)} \tag{34}$$

We must now characterize the limit $P_\infty$. The drift term $\mathcal{D}(A_t)$ acts as a driving force that only vanishes when $\sigma'(A_t) \to 0$ or the conflict disappears. Note that $\sigma'(x) = \sigma(x)(1 - \sigma(x))$, so $\sigma'(x) \to 0$ implies $\sigma(x) \to 0$ or $\sigma(x) \to 1$, i.e., $x \to -\infty$ or $x \to +\infty$.

Since the drift direction is consistently negative ($\mu < 0$), the process cannot traverse against the flow to reach $+\infty$ (which corresponds to $P = 1$). The only stable equilibrium accessible from the initialization point is the lower bound $A \to -\infty$. Therefore, the unique limit is:

$$P_\infty = 0 \tag{35}$$

This mathematically confirms that under persistent gradient conflict, the topological connection is deterministically pruned, physically decoupling the expert subspaces. □

**Corollary 1.1 (Multivariate Independence and Global Convergence).** Under Axiom 1, the physical basis $\Theta_{univ}$ is orthogonal or nearly orthogonal (low mutual coherence). This implies that the interference between dimension $j$ and dimension $k$ is negligible. Consequently, the evolution of the full topology matrix $\mathbf{A} \in \mathbb{R}^{N \times D_{base}}$ can be factorized into $N \times D_{base}$ approximately independent Markov chains. The global convergence is the product of element-wise convergences, leading the system to a discrete binary mask state $\mathbf{M} \in \{0, 1\}^{N \times D_{base}}$ that approximates the optimal masks $\{\mathbf{m}_\tau^*\}$.

### E.4. Macro-Dynamics: Thermodynamic Descent and Residual Plasticity

Finally, we link the microscopic convergence derived in Theorem 1 to the macroscopic thermodynamic evolution of the system. We prove that the system naturally evolves from a high-entropy state to a low-entropy modular state while maintaining necessary plasticity.

**Definition 4 (Structural Entropy).** We define the System's Structural Entropy $\mathcal{H}_{sys}$ as the sum of the binary entropies of all topological connections. This metric quantifies the uncertainty of the routing topology:

$$\mathcal{H}_{sys}(t) = \sum_{i=1}^{N} \sum_{j=1}^{D_{base}} H_b(\mathbf{P}_{ij}(t)) \tag{36}$$

where $H_b(p) = -p \ln p - (1 - p) \ln(1 - p)$ is the binary entropy function, with $H_b(0.5) = \ln 2$ (maximum) and $H_b(0) = H_b(1) = 0$ (minimum).

**Theorem 2 (Thermodynamic Descent to Non-Zero Entropy).** Under the dynamics of CDSP-MoE, the system entropy decreases monotonically from its maximum at initialization but stabilizes at a non-zero residual value, maintaining plasticity.

*Proof.*

**1. Entropy Reduction (Ordering Phase).** At initialization ($t = 0$), we have $\mathbf{A}_{ij} \approx 0 \implies \mathbf{P}_{ij} \approx 0.5$. Since $H_b(p)$ is strictly concave and achieves its global maximum at $p = 0.5$, the system starts at the **Maximum Entropy State** (Maximum Chaos):

$$\mathcal{H}_{sys}(0) \approx N \cdot D_{base} \cdot \ln 2 \tag{37}$$

Theorem 1 proves that under persistent conflict, probabilities are driven toward 0. Similarly, for connections where gradients align (synergy), the task gain term $\mathcal{G}$ dominates and drives probabilities toward 1. In both cases, the probability $P_{ij}(t)$ moves away from the equilibrium point 0.5.

We now compute the time derivative of entropy for a single connection:

$$\frac{dH_b(P_{ij})}{dt} = \frac{dH_b}{dP_{ij}} \cdot \frac{dP_{ij}}{dt} \tag{38}$$

$$= \ln\left(\frac{1 - P_{ij}}{P_{ij}}\right) \cdot \sigma'(A_{ij}) \frac{dA_{ij}}{dt} \tag{39}$$

Substituting Eq. (26) for $\frac{dA_{ij}}{dt}$:

$$\frac{dH_b(P_{ij})}{dt} = \ln\left(\frac{1-P_{ij}}{P_{ij}}\right)\sigma'(A_{ij}) \cdot \eta\left[-\mathcal{G} - \lambda_c\mathbf{P}_{vj}\mathcal{C}_{uv}^{(j)}\sigma'(A_{ij}) - \lambda_r\text{sgn}(A_{ij})\right] \tag{40}$$

Consider the two regimes: - **Case 1 (Conflict Dominance):** $P_{ij} \to 0$, $\ln((1-P)/P) > 0$, $\frac{dA_{ij}}{dt} < 0$, so product $< 0$. - **Case 2 (Synergy Dominance):** $P_{ij} \to 1$, $\ln((1-P)/P) < 0$, $\frac{dA_{ij}}{dt} > 0$, so product $< 0$.

In both cases, $\frac{dH_b(P_{ij})}{dt} < 0$. Summing over all connections:

$$\frac{d\mathcal{H}_{sys}}{dt} = \sum_{i,j}\frac{dH_b(P_{ij})}{dt} < 0 \tag{41}$$

This mathematically proves that the system entropy decreases monotonically, evolving from chaos to order (modularity).

**2. Residual Plasticity (Stationary Phase).** The SGD noise $\xi_t$ acts as a thermal bath with effective temperature $T_{eff} \propto \eta \cdot \text{Var}(\xi_t)$. According to the Fokker-Planck equation for Langevin dynamics (Risken, 1996), the probability distribution $\rho_t(\mathbf{A})$ converges to a stationary Boltzmann distribution:

$$\lim_{t\to\infty}\rho_t(\mathbf{A}) \propto \exp\left(-\frac{\mathcal{H}(\mathbf{A})}{T_{eff}}\right) \tag{42}$$

The entropy of this stationary state is strictly positive:

$$S_{min} = -\int \rho_\infty(\mathbf{A})\ln\rho_\infty(\mathbf{A})d\mathbf{A} \tag{43}$$

$$= \frac{\mathbb{E}_{\rho_\infty}[\mathcal{H}]}{T_{eff}} + \ln Z > 0 \tag{44}$$

where $Z = \int \exp(-\mathcal{H}(\mathbf{A})/T_{eff})d\mathbf{A}$ is the partition function.

This result is crucial: it ensures that the final topology retains **Residual Plasticity**. The connections are not "frozen" at exactly 0 or 1, but fluctuate slightly around the optimal mask $\mathbf{m}^*$ driven by the "thermal noise" of the data. The regularization term $\lambda_r\|\mathbf{A}\|_1$ contributes to this thermal effect, preventing premature freezing into suboptimal configurations. $\qquad\square$

**Empirical Correspondence:** This theoretical framework directly explains the experimental observations in Section 4.3.1 of the main paper. The "oligarchic" structure in Fig. 2 corresponds to the low-entropy stationary distribution $\rho_\infty$, where a subset of experts (E5, E7, E2, E0) have strong connections ($P \approx 1$) while others are pruned ($P \approx 0$). The residual plasticity $S_{min} > 0$ explains why the system maintains adaptability even after convergence, as observed in the "Uncertainty Drift" phenomenon in Fig. 3d.

**Summary:** We have constructed a complete theoretical framework demonstrating that: 1. Gradient conflict is inevitable under the Universal Weight Subspace Hypothesis (Lemma 1). 2. Conflict drives topological connections to be pruned, leading to subspace decoupling (Theorem 1). 3. The system evolves from high entropy to low entropy, achieving modularity (Theorem 2). 4. Residual plasticity is maintained due to SGD noise and regularization, preventing brittle collapse.

This analysis provides a rigorous foundation for the emergent modularity observed in CDSP-MoE, positioning it as a principled approach to autonomous structure learning in neural networks.

