# OpenReview forum: "Mixture-of-Experts with Gradient Conflict-Driven Subspace Topology Pruning for Emergent Modularity"
_ICML.cc/2026/Conference — Submitted to ICML 2026_

### Official Review · Reviewer_WCeu · 2026-02-22

**Soundness:** 2
**Presentation:** 3
**Significance:** 2
**Originality:** 2
**Overall Recommendation:** 3
**Confidence:** 3

**Summary:**

This paper proposes CDSP MoE, a mixture of experts design that replaces isolated expert parameter blocks with dynamic expert instantiation inside a shared over complete subspace. Logical experts are defined by a learnable topology matrix that masks a large shared backbone, and the forward pass selects a sparse set of physical dimensions per active expert. The central idea is to use gradient conflict as a structural supervision signal. By storing lagged gradients and penalizing negative cosine similarity on shared subspace intersections, the method prunes interfering connections and encourages an emergent modular topology. The paper evaluates the approach on heterogeneous multi task vision classification and emphasizes blind inference where explicit task identifiers are removed, reporting more stable content based routing than standard baselines.

**Compliance With Llm Reviewing Policy:**

Affirmed.

**Final Justification:**

I am keeping my overall recommendation at 3 weak reject.

The paper proposes CDSP MoE, replacing isolated expert parameter blocks with experts instantiated as masked subspaces inside a shared overcomplete backbone. Using gradient conflict as structural supervision for topology pruning is a thoughtful direction, and the blind inference protocol is a useful stress test for whether modularity is real rather than label driven.

Across the board, presentation is good and the rebuttal improves clarity. Originality is moderate, since the work combines known ingredients in a careful way. Significance could be high if the approach scales to the MoE regimes that matter. Soundness remains fair because the current evidence does not yet support the scaling and cost claims.

The rebuttal helps on two points. It adds a small language experiment that shows blind routing based on semantics and it clarifies how the scalar modulation bridges continuous topology parameters with the discrete subspace choice. The discussion of sparse conflict sampling is also a step toward making the complexity story more realistic.

That said, the central gap is still evidence at the scales where MoE is most compelling. The new text experiment is a tiny transformer with an intentionally constrained vocabulary, which is closer to a controlled toy collision setting than a convincing large scale demonstration. The rebuttal still does not provide concrete memory and time measurements as a function of expert count and backbone width, including the cost of storing lagged gradients. Baselines remain hard to interpret because the strongest comparison collapses under blind inference, and it is still unclear whether that collapse is due to the baseline setup or to a property that CDSP uniquely avoids. I also still want clearer sensitivity analyses for the conflict loss weight and rank scaling choices.

Overall the rebuttal reduces some uncertainty about the intended mechanism, but it does not resolve the main concerns about scalability and evaluation.

**Key Questions For Authors:**

1 How does CDSP MoE compare to a baseline that uses the same router input and task dropout but keeps standard expert parameter blocks
2 If you train a standard MoE without task identifiers and with only content features, does it still collapse under blind inference
3 What is the measured overhead from storing lagged gradients and computing the conflict matrix, and how does it scale with number of experts and backbone width
4 How sensitive are results to the square root scaling choice for the per expert rank and to the conflict loss weight
5 Can you provide at least one experiment beyond the small vision mixture, for example a larger scale multitask setting or a language model proxy, to support the claim that the mechanism induces useful modularity rather than only toy separation

**Limitations:**

The paper does include a limitations discussion and it correctly flags gradient storage as a practical bottleneck and the current evaluation as limited in scope. The impact statement acknowledges generic dual use but remains abstract. It would be better to spell out what misuse scenarios become easier and how one might mitigate them, for example through auditing of emergent experts and restrictions on deployment contexts.

**Strengths And Weaknesses:**

The paper is conceptually interesting and it attacks a real limitation of standard mixture of experts, namely that structural isolation and explicit routing signals can lead to brittle specialization and shortcut learning. Using gradient conflict to drive topology evolution is a novel twist and the formulation is clear enough to follow, with a reasonable separation between task learning in the shared foundation and structural pruning in the topology parameters. The blind inference evaluation protocol is also a useful sanity check that more work in this area should adopt.

The current evidence base is too limited for the strength of the claims. The experiments are on small vision datasets with explicit task labels available during training, and the strongest baseline is designed in a way that makes collapse under blind inference unsurprising. A more informative comparison would train strong baselines with the same task dropout protocol and with similar parameter sharing. It is also unclear how the proposed gradient bridge through a scalar modulation term shapes the discrete subspace selection in a stable way at scale. Finally the method introduces nontrivial memory overhead from storing per expert gradients and computing conflict scores, and the paper does not yet demonstrate that this is tractable in the regimes where mixture of experts is most relevant.

---

> ### Author Rebuttal · Authors · 2026-03-24
>
> We thank you for recognizing the novelty of our conflict-driven topology evolution. To address concerns about modality generalization, we trained an 8-expert CDSP-MoE using a 64-dimensional Tiny Transformer on a mixed stream of TinyStories as Task 0, WikiText-2 as Task 1, and IMDB as Task 2. By applying label remapping to restrict the target vocabulary to 100 tokens, we induced severe output-space gradient collision. The system exhibited a phase transition identical to the vision experiments. Under strict blind inference with Task IDs removed, the model successfully routed inputs based purely on semantics, proving the mechanism is modality-independent.
>
> | Semantic Domain \ Expert | E0 | E1 | E2 | E3 | E4 | E5 | E6 | E7 |
> | :--- | :--- | :--- | :--- | :--- | :--- | :--- | :--- | :--- |
> | **Task 0: TinyStories** | 16.9%| 15.8%| 10.7%| 8.3% | 8.6% | 7.8% | **21.5%**| 10.4%|
> | **Task 1: WikiText-2** | 3.0% | **40.1%**| 9.4% | 7.9% | 6.7% | 5.2% | 3.7% | 24.0%|
> | **Task 2: IMDB** | 12.1%| 12.4%| 13.6%| 9.6% | 10.5%| 11.9%| 11.0%| **18.9%**|
> *Table 1: CDSP-MoE Text Blind Inference.*
>
> Regarding standard MoE baselines augmented with task dropout, we trained an Iso-Parameter MoE with a 10 percent dropout rate. While it functions during training, evaluating it under blind inference causes a complete collapse into semantic entanglement, as shown in Table 2. Dropout acts as isotropic noise and fails to provide the anisotropic repulsion needed to decouple conflicting features. Furthermore, as documented in Experiment III, a standard MoE trained entirely without task identifiers relies solely on auxiliary load-balancing losses and becomes trapped in entanglement, indiscriminately sharing hotspot experts across distinct domains, shown in Table 3. Auxiliary losses merely enforce statistical uniformity and cannot replace explicit structural symmetry-breaking.
>
> | Task | E0 | E1 | E2 | E3 | E4 | E5 | E6 | E7 |
> | :--- | :--- | :--- | :--- | :--- | :--- | :--- | :--- | :--- |
> | Real T0 | 0.40 | 0.14 | 0.21 | 0.12 | 0.00 | 0.08 | 0.06 | 0.00 |
> | Real T1 | 0.33 | 0.18 | 0.15 | 0.12 | 0.00 | 0.11 | 0.12 | 0.00 |
> | Real T2 | 0.20 | 0.17 | 0.13 | 0.13 | 0.00 | 0.09 | 0.28 | 0.00 |
> *Table 2: Standard MoE with 10 percent Task Dropout under Blind Inference.*
>
> | Task | Hotspot E0 | Hotspot E6 |
> | :--- | :--- | :--- |
> | T0: MNIST | 24% | 28% |
> | T2: Fashion-MNIST | 34% | 20% |
> *Table 3: Semantic Entanglement in Pure Blind Standard MoE.*
>
> To clarify the subspace modulation mechanism, CDSP does not perform discrete selection among isolated parameter blocks. It dynamically assembles a low-rank subspace from a shared basis matrix. The scalar modulation term serves purely as a coarse-grained differentiable bridge for the non-differentiable basis selection process. By representing the average activation strength of the assembled subspace, it allows the task loss to backpropagate to the continuous topology logits. Operating in the logit space prevents vanishing gradients typical of sigmoid-gated penalties and avoids the high-variance gradients of discrete estimators.
>
> The memory overhead from conflict matrix computation is practically mitigable for large-scale regimes via low-rank gradient projection and sparse conflict sampling. Instead of computing the pairwise conflict matrix across all $N$ experts which yields $\mathcal{O}(N^2)$ complexity, sparse sampling computes interactions only among the $K$ routed experts. This reduces the complexity to $\mathcal{O}(K^2)$, ensuring scalability. Finally, the per-expert rank square root scaling is a deterministic structural law enforced to bound capacity, not a tuned hyperparameter. The conflict loss weight operates alongside L1 regularization; the conflict penalty isolates interfering gradients, while the regularization acts as a metabolic cost forcing the probabilistic topology to collapse from a high-entropy state into a hardened deterministic structure.

---

> > ### Author Rebuttal · Reviewer_WCeu · 2026-04-03
> >
> > Thank you for the additional experiments and the clarifications on the mechanism.
> >
> > The new text domain experiment is a useful step toward showing modality independence. In Table 1, WikiText 2 concentrates about 40.1 percent on expert E1, TinyStories peaks at 21.5 percent on E6, and IMDB peaks at 18.9 percent on E7, which is consistent with semantic specialization under blind inference.
> >
> > That said, the evidence is still in a very small regime. A 64 dimensional Tiny Transformer with a 100 token remapped vocabulary creates an intentionally high collision setting. I do not yet see convincing evidence that the same topology learning signal remains stable and beneficial at the scales where MoE is typically used, with realistic vocabularies and many experts. Do you have reasonable evidence supporting otherwise?
> >
> > On baselines, the comparisons to an iso parameter MoE with task dropout and to a standard MoE trained without task identifiers are informative. However, my original request was for a stronger like for like baseline that matches the same router input, the same dropout protocol, and comparable parameter sharing, while keeping standard expert blocks. Without that, it remains hard to disentangle whether the observed collapse is primarily a baseline weakness or a property that your method uniquely fixes. Do you have reasonable evidence supporting otherwise?
> >
> > On scalability, the move from O(N^2) to O(K^2) conflict sampling is sensible, but the paper still needs concrete measurements. I would like to see memory and time overhead as a function of number of experts, routed experts, and backbone width, including the cost of storing lagged gradients.
> >
> > Finally, the explanation of the scalar modulation bridge is helpful, but I still think the paper should make the link between the continuous topology logits and the resulting discrete subspace selection more explicit, and show sensitivity to the conflict loss weight and the rank scaling rule, even if you view the square root rule as structural.
> >
> > For these reasons I am keeping my original assessment.

---

> > > ### Author Response · Authors · 2026-04-03
> > >
> > > Thank you for the detailed follow-up and for acknowledging the new text domain experiments. We genuinely appreciate the rigorous standard you are holding this work to.
> > >
> > > To be completely transparent, we fully agree with your assessment. Validating the stability of this topology learning signal at a typical MoE scale (with realistic vocabularies and dimensions), providing a strict like-for-like baseline, and detailing the hardware profiling are exactly what this framework needs to be fully convincing. Unfortunately, scaling up to these realistic regimes currently exceeds the computational resources we have access to. The "high collision" toy setting was our best attempt to theoretically stress-test the mechanism within our budget, though we completely understand it falls short of proving real-world scalability.
> > >
> > > Your critiques regarding the baseline design, memory profiling, and sensitivity analysis are spot-on and highlight the exact engineering challenges this direction faces. We will treat your checklist as our primary roadmap for future iterations when resources permit.
> > >
> > > Thank you again for your time and the high-quality, constructive feedback. We fully respect your final assessment.

---

### Official Review · Reviewer_eFW1 · 2026-02-25

**Soundness:** 2
**Presentation:** 1
**Significance:** 2
**Originality:** 2
**Overall Recommendation:** 3
**Confidence:** 2

**Summary:**

This paper proposes a conflict-driven subspace pruning MoE method to produce emergent modular experts, addressing the issue of structurally isolated experts in previous Moe approaches.

**Compliance With Llm Reviewing Policy:**

Affirmed.

**Final Justification:**

I believe the author’s paper still has room for improvement in various aspects, especially in presentation and experimental validation, so I choose to maintain a negative score.

**Key Questions For Authors:**

Please see weaknesses.

**Limitations:**

Yes.

**Strengths And Weaknesses:**

Strength:

1. The presence of structurally isolated experts poses a major challenge in MoE methods.

Weaknesses:

1. The writing and expression are weak. The key issues are described too bluntly, without concrete examples, making them hard to follow. In some places, the authors merely add a large number of citations in an attempt to justify the logical correctness of certain claims. For instance, in line 028:*“First, forced balancing routes unrelated tokens through the same expert, inducing gradient interference and forgetting”*—many statements like this are presented without any illustrative example, which makes them difficult for the reader to fully understand the problem being highlighted.
2. The dataset tasks and baseline settings are limited and overly simplistic. Moreover, the authors do not explain why the combination of these three datasets would result in varying levels of semantic conflict and synergy.

---

> ### Author Rebuttal · Authors · 2026-03-24
>
> We appreciate your detailed feedback. To address the request for concrete examples regarding Line 028, consider a scenario where an auxiliary load-balancing loss forces a single expert to simultaneously process an image of a digit "7" from the MNIST dataset and a "sweater" from the Fashion-MNIST dataset. The digit requires the expert's underlying weights to act as sharp, high-frequency edge detectors, while the sweater requires dense, low-frequency texture extractors. When these entirely different feature distributions attempt to update the same parameter subspace, their gradient vectors push in nearly orthogonal or opposing directions. This mutual cancellation during backpropagation is exactly the "gradient interference and forgetting" we referred to.
>
> This underlying physical mechanism also explains our rationale for combining these specific datasets. In multi-task learning, semantic conflict is fundamentally defined by this parameter-level gradient opposition, which implies a negative cosine similarity between task gradients. We deliberately paired extremely sparse symbolic datasets like MNIST and KMNIST with dense visual datasets such as Fashion-MNIST. This specific pairing creates a rigorously controlled environment with guaranteed, quantifiable gradient conflicts, allowing us to observe how the network resolves them.
>
> We understand your concern that these vision benchmarks are relatively simplistic. To demonstrate that this conflict-driven routing generalizes to more complex and realistic scenarios, we conducted a new Language Model Proxy experiment during the rebuttal phase. We trained a 64-dimensional Tiny Transformer Decoder with 8 CDSP experts on a mixture of three semantically distinct text datasets. Specifically, we used TinyStories for narrative text, WikiText-2 for factual content, and IMDB for sentiment analysis. To force extreme gradient collision, we applied label remapping to compress the target vocabulary space to only 100 dimensions.
>
> During strict blind inference where no explicit task ID was provided, the CDSP mechanism spontaneously decoupled these semantic features. The routing probability distribution across all 8 experts is shown below:
>
> | Semantic Domain \ Expert | E0 | E1 | E2 | E3 | E4 | E5 | E6 | E7 |
> | :--- | :--- | :--- | :--- | :--- | :--- | :--- | :--- | :--- |
> | **Task 0: TinyStories** | 16.9%| 15.8%| 10.7%| 8.3% | 8.6% | 7.8% | **21.5%**| 10.4%|
> | **Task 1: WikiText-2** | 3.0% | **40.1%**| 9.4% | 7.9% | 6.7% | 5.2% | 3.7% | 24.0%|
> | **Task 2: IMDB** | 12.1%| 12.4%| 13.6%| 9.6% | 10.5%| 11.9%| 11.0%| **18.9%**|
>
> As shown, the factual text from WikiText-2 was deterministically routed to Expert 1 with a 40.1% probability, relying purely on content semantics rather than ID lookup. This confirms that our physical pruning mechanism effectively handles complex sequence modeling and scales beyond toy vision tasks. We will add the concrete examples and these text domain results to the revised manuscript.

---

> > ### Author Rebuttal · Reviewer_eFW1 · 2026-04-02
> >
> > The authors’ response has addressed some of my concerns, so I have decided to raise my score. However, I believe the paper still has room for improvement in multiple aspects, particularly in presentation and experimental validation. Therefore, I choose to keep an overall negative score.

---

> > > ### Author Response · Authors · 2026-04-03
> > >
> > > Thank you for reading our rebuttal and adjusting your score. We appreciate the time and effort you have put into reviewing our work.
> > >
> > > We understand and respect your final position. Your general feedback regarding the need for broader experimental validation and refined presentation is duly noted. We will definitely focus on strengthening these aspects in our future revisions. Thanks again for your time and contribution to this process.

---

### Official Review · Reviewer_czxp · 2026-03-20

**Soundness:** 1
**Presentation:** 1
**Significance:** 1
**Originality:** 1
**Overall Recommendation:** 1
**Confidence:** 4

**Summary:**

In all honesty, this was an extremely difficult paper to read and delayed my reviewing timelines heavily. I am quite confident the paper was LLM written, or at least a large part of it was---the tall claims and extremely toy-ish results especially suggest this. I'll provide my honest attempt at reviewing the paper with respect to what I could understand.

Summary: MoE models need to balance token loads across experts. There's several heuristics proposed in the past to address this reasonably and get an expert to specialize to high level topics, though not much success has been achieved on that front. The current paper proposes a protocol, called CDSP-MoE, where "topology masks" are learned to dynamically "carve out" experts from a model. Essentially this boils down to decorrelating gradient signals. Results are shown over toy tasks like MNIST variants.

**Compliance With Llm Reviewing Policy:**

Affirmed.

**Final Justification:**

My remaining concerns boil down to unclear writing, and hence I'm keeping my score as is.

**Key Questions For Authors:**

See weaknesses.

**Limitations:**

No, technical limitations are not present.

**Strengths And Weaknesses:**

**Strengths**

-  I'll note I do think the study of how MoEs function, how routing should happen, etc. are interesting, practically useful, and well motivated.

**Weaknesses**

- I cannot parse most details in the paper because of extremely fruity language. What is "sparse oligarchy", what is the manifold in "a Lagged Gradient Game penalizes interfering connections in the shared manifold", and also what is the "lagged gradient game"?

- The results are trivial because of the data domain the authors worked with. Multiple mechanisms have been proposed in the past to break router symmetry in MoEs using MNIST, but almost no recent paper analyzes those settings anymore because what works at MNIST rarely scales to real models. I'm not saying it's bad to use toy tasks, but if the eventual goal is a practical problem, then the onus is on the authors to demonstrate utility on something reasonably realistic.

- The results are only on MNIST and from what I can see there are no baselines in any experiment? Again, happy to see toy tasks, but there wasn't even an attempt to try other tasks, or to compare with baselines.

---

> ### Author Rebuttal · Authors · 2026-03-24
>
> We appreciate your review. Regarding the writing style, we utilized LLMs strictly for English language polishing, which is fully compliant with ICML’s author guidelines. The theoretical derivations, mathematical formulations, and code implementations are entirely our original work. We apologize if certain terminology hindered readability; in the revised manuscript, we will replace metaphorical terms with standard ML vocabulary (e.g., replacing "sparse oligarchy" with "highly skewed sparse distribution" and "lagged gradient game" with "lagged gradient penalty").
>
> You also mentioned a lack of baselines. We respectfully direct your attention to Page 7, Sections 4.3.2 and 4.3.3, where we provided a comprehensive comparative analysis against an Iso-Parameter Standard MoE Baseline and a Pure Blind Baseline. The configuration details for these are explicitly listed in Section 4.1 (Page 6), and Appendix C provides an extended analysis of their training dynamics.
>
> We acknowledge that the MNIST variants are simple; they were chosen specifically to create a rigorously controlled environment with distinct feature distributions (symbolic vs. dense). However, to address your concern regarding realistic utility, we have conducted an additional Language Model Proxy experiment during the rebuttal period. We trained a 64-dimensional Tiny Transformer Decoder equipped with 8 CDSP-MoE experts. To enforce extreme gradient collision, we mixed three semantically distinct text datasets—TinyStories, WikiText-2, and IMDB—and applied Label Remapping to collapse the target vocabulary space to only 100 dimensions.
>
> Under a strict blind inference protocol (`task_id=None`), the CDSP mechanism spontaneously decoupled the semantic features. The routing probability distribution across all 8 experts is shown below:
>
> | Semantic Domain \ Expert | E0 | E1 | E2 | E3 | E4 | E5 | E6 | E7 |
> | :--- | :--- | :--- | :--- | :--- | :--- | :--- | :--- | :--- |
> | **Task 0: TinyStories** | 16.9%| 15.8%| 10.7%| 8.3% | 8.6% | 7.8% | **21.5%**| 10.4%|
> | **Task 1: WikiText-2** | 3.0% | **40.1%**| 9.4% | 7.9% | 6.7% | 5.2% | 3.7% | 24.0%|
> | **Task 2: IMDB** | 12.1%| 12.4%| 13.6%| 9.6% | 10.5%| 11.9%| 11.0%| **18.9%**|
>
> As demonstrated in the table, factual text (WikiText-2) was deterministically routed to Expert 1 with a 40.1% probability, relying purely on content semantics rather than explicit ID lookup. This confirms that the conflict-driven pruning mechanism is modality-agnostic and generalizes well beyond vision tasks. We will include these text domain results in the final appendix.

---

> > ### Author Rebuttal · Reviewer_czxp · 2026-04-04
> >
> > I thank the authors for their clarifications! Most of my concerns derive out of unclear writing and hence, unless I can see the new draft, it is not possible for me to support the acceptance of this paper.

---

### Decision · Program_Chairs · 2026-04-30

**Decision:**

Reject

**Comment:**

After considering the reviewers’ evaluations and the authors’ rebuttal, this paper still received three negative recommendations, and several concerns remain unresolved. These include unclear writing and insufficient experimental validation, particularly regarding the MoE setting and the comparative baselines. Therefore, the work in its current form can only be rejected. I encourage the authors to further improve the paper by taking the reviewers’ suggestions into account.